# Estimation of Children’s Soil and Dust Ingestion Rates and Health Risk at E-Waste Dismantling Area

**DOI:** 10.3390/ijerph19127332

**Published:** 2022-06-15

**Authors:** Yan Yang, Mengdi Zhang, Haojia Chen, Zenghua Qi, Chengcheng Liu, Qiang Chen, Tao Long

**Affiliations:** 1School of Environmental Science and Engineering, Institute of Environmental Health and Pollution Control, Guangdong University of Technology, Guangzhou 510006, China; mengdizhang0120@163.com (M.Z.); chenhaojia_gdut@163.com (H.C.); qizenghua@126.com (Z.Q.); 2Chemistry and Chemical Engineering Guangdong Laboratory, Shantou 515041, China; 3Synergy Innovation Institute of GDUT, Shantou 515041, China; 13610227100@163.com; 4State Environmental Protection Key Laboratory of Soil Environmental Management and Pollution Control, Nanjing Institute of Environmental Sciences, Ministry of Ecology and Environment of China, Nanjing 210042, China; chenqiang@nies.org (Q.C.); longtao@nies.org (T.L.)

**Keywords:** e-waste, heavy metal pollution, children, soil and dust ingestion rates, health risk assessment

## Abstract

Due to environmental health concerns, exposure to heavy metals and related adverse effects in electronic waste (e-waste) dismantling areas have attracted considerable interest in the recent years. However, little information is available about the Soil/Dust Ingestion Rates (SIR) of heavy metals for children living in such sites. This study estimated the soil ingestion of 66 children from e-waste disassembly areas by collecting and analyzing selected tracer elements in matched samples of their consumed food, feces, and urine, as well as soil samples from their play areas. The concentrations of tracer elements (including Al, Ba, Ce, Mn, Sc, Ti, Y, and V) in these samples were analyzed. The SIR was estimated to be 148.3 mg/day (median) and 383.3 mg/day (95th percentile) based on the Best Tracer Method (BTM). These values are somewhat higher than those observed in America, Canada, and other parts of China. Health risk assessments showed that Cr presented the greatest carcinogenic risk, at more than 10^−6^ in this typical polluted area, while As was second. These findings provide important insights into the exposure risks of heavy metals in e-waste dismantling sites and emphasize the health risk caused by Cr and As.

## 1. Introduction

The potentially harmful environmental and human health effects of primitive electronic-waste (e-waste) recycling processes, including manual disassembly, roasting, acid leaching, and open burning, have caused concern around the world, particularly in rapidly industrializing and urbanizing developing countries such as China, India, and Vietnam [1,2]. Notably, heavy metal pollution is ubiquitous in the environment and bodies of people living near e-waste disposal sites [3,4]. Heavy metal elements accumulate in the human body and interfere with the human endocrine system [5], damage the body’s cardiovascular and nervous systems [6,7], and can even lead to cancer [8]. It has been reported that the concentration of Cu was about three times higher than the Grade II guideline level (Soil Environmental Quality Standard, GB 15618–1995) in an e-waste disassembly area in China, and the Cu concentrations of soils from dumping, burning, and acid leaching sites were found to be 10, 40, and 60 higher, respectively, than the Grade II level. It was also reported that concentrations of blood lead, cadmium, and lead in meconium were higher in children and newborns living in e-waste disassembly areas than in neighboring areas [9,10]. In addition, children also have higher frequencies of mouthing behaviors, higher ingestion rates, lower body weights, and are more vulnerable to toxic substances, as compared to adults [11]. Therefore, it is necessary to assess the exposure levels and health risk caused by hand-to-mouth/object-to-mouth transfer that children face due to heavy metals in e-waste disassembly areas.

To more accurately assess the health risks that heavy metals post to children resulting from hand-to-mouth/object-to-mouth exposure, the Soil/Dust Ingestion Rate (SIR) is an important factor when estimating the risks caused when children are exposed to pollutants that are prone to binding to soils, such as heavy metals [12]. An estimation of the daily SIR of children from Gansu Province via hand-to-mouth contact showed that kindergarten and primary school children ingested 7.73 and 6.61 mg/day, respectively [13]. Lin et al. presented the first large-scale study of SIR for 177 Chinese children and recommended SIR values for the general population of Chinese children (from 2.5 to 12 years old): 52 mg/day was the central tendency and 217 mg/day was the upper percentile [14]. The SIR of children from 6 to 71 months old in the United States was found to be 85 mg/day [15]. This value was subsequently set as the recommended SIR value for children under one year of age by the USEPA; Chinese guidelines reference this. Due to various factors, including different living behaviors, different SIR values are observed in different regions. However, the coefficients used to compute SIR values are often based on the results of studies conducted in the United States, and directly applying these coefficients may lead to errors in studies conducted elsewhere. Therefore, independent SIR determinations should be conducted in individual districts. The need for accurate regional SIR data is enhanced by the fact that some areas, such as e-waste dismantling sites, have very high levels of heavy metal contamination in soil and dust. Consequently, reliable information on exposure factors and SIR in such regions is urgently needed to enable an accurate assessment of children’s health risks.

At present, three experimental methods are available to estimate the SIR of children: the activity pattern-based methodology [16,17,18], the biokinetic modeling methodology [19,20], and the tracer element methodology [21,22,23]. The tracer element methodology, which is suitable for all situations and based on accurate experimental data, has been widely used to determine SIR since 1980 [14,24,25]. To calculate SIR (US guidelines), this method analyzes the concentration of tracer elements in the soil to which children are exposed, the children’s intake of food, their excreted feces and urine, and the tracer element contents in the children’s food, feces, and urine. However, because the lowest estimated SIR determined for a given tracer element will always be greater than the actual SIR of the human body, researchers developed the Limitation Tracer Method (LTM), which defines the soil intake as the lowest of the individual estimated values for a set of tracer elements [13,26]. Then, to improve the accuracy of the LTM which does not look at ingestion traces in food or medicine, the Best Tracer Method (BTM) was put forward to obtain a child’s SIR [25]. At present the BTM is the most suitable method of accurately quantifying SIR [14,27].

Therefore, this study focuses on estimating the children’s SIRs in e-waste disassembly areas by collecting and analyzing selected tracer elements in matched samples of their food, feces, and urine, as well as soil samples from their play areas, to further assess the health risk to children in e-waste dismantling areas.

## 2. Materials and Methods

### 2.1. Study Site and Sampling

The study examined a population sample of 66 children, most of whom lived in an e-waste dismantling area in South China. The e-waste dismantling area we studied is an E-Waste Recycling Town located in South China, where a possible human body burden and health consequences due to heavy metals exposure have been reported [10]. The ages, heights, and weights of the participating children and some associated descriptive statistics are presented in Appendix A. Ages ranging from 3 to 17 years old, with a median of 9 years old, were chosen. The median (maximum to minimum) weight and height were 19.0 kg (7.0–72.0 kg) and 113.0 cm (74.0–160.0 cm), respectively.

The children mainly came from a full-day school and kindergarten. Samples of their food, urine, and feces were collected daily by their parents, guardians, and teachers. Each participant was forbidden to take any drugs during the sampling period to reduce errors. It is generally assumed that there is a lag time of a 28-h from the ingestion of food and soil to the resulting fecal and urinary output [28]. Sample collections followed the USEPA recommendation of a 28-h period from food to feces and urine (for example, food collection from day-1 morning (approximately at 07:00 h, including breakfast) to day-2 morning (but not day-2 breakfast) and feces collection from day-2 morning (approximately at 07:00) to day-3 noon (approximately 11:00)). Therefore, the sample collection corresponding to one day lasted 52 h (collecting food in the first 24 h and collecting feces and urine after 28 h).

A “duplicate plate” method (two identical meals were prepared: one was for the subjects to eat and the other was mixed into a food sample and weighed for laboratory determination) was used to collect food samples, including breakfast, lunch, and children’s dinner. Food samples (*n* = 66) were weighed before being lyophilized and then crushed. All feces (*n* = 62) and urine (*n* = 64) for each subject were collected daily using pre-labeled and pre-weighed portable sample containers, respectively. When the collection was completed, the samples were taken back to the laboratory and stored first in the refrigerator (−20 °C). Then, the feces were freeze-dried (under vacuum conditions, the vacuum freeze drier temperature ranges from −40 °C to −50 °C for 48 h) after measuring the weight with a vacuum freeze-dryer. The urine was stored after measuring the volume. Topsoil (*n* = 5) and dust (*n* = 3) samples were collected from campuses and green spaces where children generally play, respectively. Dust was collected indoors or outdoors by cleaning dust from areas such as tables and windowsills. A total of 20 g of soil and as much dust (5–20 g) as possible were collected during the sampling process. After air-drying, all samples were crushed with a ceramic mortar and pestle and then passed through a 0.25-mm sieve. This sampling method was also used to collect the 46 soil samples that were used to measure the heavy metals, including 24 residential areas and 20 park green areas. A combined assessment of local heavy metal pollution levels and children’s SIR enabled accurate assessment of the health risks facing children due to local soil contamination and the associated heavy metal intake. All samples were collected in 2019 and stored at −20 °C for later processing and analysis.

### 2.2. Sample Preparation and Instrumental Analysis

Feces samples and food samples were pretreated in the same way. Dried samples (1 g) were digested to evaporate at low temperatures of 55 °C on a heating plate with 3 mL concentrated nitric acid, 3 mL hydrogen fluoride, and 1 mL perchloric acid (HNO_3_–HF–HClO_4_). The digestion process was repeated until the sample became sticky. Crushed evaporated samples were microwaved with 2 mL concentrated nitric acid and 3 mL hydrogen peroxide (HNO_3_–H_2_O_2_). Digestion was performed at 120 °C for 5 min, then 160 °C for 5 min, and finally 180 °C for 15 min. The digested product was diluted to 30 mL with ultrapure water and then stored at 4 °C. The supernatant was extracted and analyzed by High-Resolution Inductively Coupled Plasma Mass Spectrometry (HR–ICP–MS, Nu Attom, North Wales, UK) to determine Mn, Al, Ba, Ti, Ce, V, Sc, and Y.

Urine samples (15 mL) were placed in a digestion tube and digested by microwaving with 2 mL of H_2_O_2_ and 3 mL of concentrated HNO_3_. The conditions of microwave digestion are the same as mentioned above. Finally, the sample after digestion was analyzed by HR–ICP–MS.

Dried soil and dust samples (0.5 g) were digested the same as feces and food samples on a heating plate by HNO_3_–HF–HClO_4_. Then, samples were microwaved with HNO_3_–H_2_O_2_. The supernatant was analyzed for Al, Ba, Mn, Ti, and V by Inductively Coupled Plasma Optical Emission Spectroscopy (ICP–OES, Spectra Arcos SOP, Kleve, Germany), which is a method of atomic emission spectroscopy analysis using a light source that generates plasma discharge through high-frequency inductive coupling and ICP–MS for Ce, Sc, and Y. In addition, dried soil samples from living spaces (0.5 g) were digested with the same pre-treatment and analyzed by ICP–MS for Pb, As, Cr, Cu, Ni, Cd, and Zn. The standard concentration curve was used to determine the sample concentration established by the heavy metal standard (all standards were from The Nonferrous Metals Society of China).

### 2.3. QA/QC Method for Analytical Data

The accuracy of the method used to analyze tracer elements in the collected food, feces, urine and soil samples was tested by using the same method to detect the same elements in substrate mixed standards, while certified reference materials replaced dust and soil substrates. Analysis of the certified reference materials with the same elements was used to show the accuracy of the method used for the analysis of heavy metals in soil samples. The tracer elements recoveries from soil and dust, food, and fecal and urine samples ranged from 74.2% to 102.3%, 83.4% to 108%, and 84.6% to 110%, respectively, for Al, Ba, Mn, Ti, and V analyzed by ICP–OES. The recovery of tracer elements from soil and dust, food, and fecal and urine samples ranged from 79.3% to 97.3%, 82.1% to 110%, and 81.1% to 101%, respectively, for Ce, Sc, and Y analyzed by ICP-MS. Furthermore, the recovery of Pb, As, Cr, Cu, Ni, Cd, and Zn analyzed by ICP-MS from soil and dust ranged from 71.3% to 93.9%. All reagents used in the analysis were of high purity. The experimental water was ultra-pure, and all the glassware was soaked for more than 36 h in 10% nitric acid before use. The results showed that the recovery of various heavy metal elements ranged from 95% to 105%. The Ce, Sc, and Y concentrations in the supernatant were defined as half of their detection limits, because all of them were under their detection limits, which were 0.001 ng/mL, 0.001 ng/mL and 0.002 ng/mL, respectively).

### 2.4. SIR Estimates and Best Mass-Tracer Method

Based on the experimental data, the daily SIR of each participating child was calculated using the following expression. After caluclating SIR values for each element, the best mass-tracer method was used to obtain a more representative SIR value.
(1)$$SIR=\frac{\left[\left(W_{feces}\times C_{feces}+V_{urine}\times C_{urine}\right)-\left(W_{food}\times C_{food}\right)\right]}{C_{soil/dust}}$$
where *SIR* is the SIR (mg/day) each day over the study period for each kid. The dry weight of feces is *W_feces_* (kg/day), the concentration of tracer elements in feces is *C_feces_* (mg/kg), the urine volume is *V_urine_* (mL/day), the concentration of tracer elements in urine is Curine (μg/mL), the weight of food consumed is *W_food_* (kg/day), the concentration of tracer elements in food is *C_food_* (mg/kg), and the concentration of tracer elements in soil and dust is *C_soil/dust_* (mg/kg) [28]. Data for *W_feces_*, *V_urine_*, and *W_food_* are shown in Appendix A, while data on *C_feces_*, *C_urine_*, and *C_food_* are shown in Appendix A.

### 2.5. Human Health Risk Assessment

The Average Daily Dose (*ADD*, kg kg^−1^ bodyweight day^−1^), which is used to assess the health risk due to intake of toxic materials [28], can be calculated with the following equation:(2)$$ADD=\left(C\times SIR\times EF\times \frac{ED}{BW}\times AT\right)\times CF$$
where the heavy metal concentration in soil/indoor dust (kg/kg) is *C*, the soil and dust ingestion rate is *SIR* (mg/day); the exposure frequency is *EF* (days/years) and is taken as 350 d/y; the exposure duration is *ED* (year) and is taken as 6 years for per child; the body weight is *BW* (kg); the average time is *AT* (day), for non-carcinogenic effect, which was taken to be 2190 days (for carcinogenic effect, *AT* is 27,740 days); the unit conversion factor of 10^−6^ is *CF* [28].

The Hazard Quotients (*HQ*) are the ratio of daily intake dose of pollutants to reference dose, which is used to characterize the levels of human exposure to non-carcinogenic contaminants through a single pathway which represents the level of non-carcinogenic risk. This is calculated as follows:(3)HQ=ADD/RfD
where the estimated maximum permissible dose to humans via oral ingestion exposure is *RfD* (mg/(kg·d)); in this study, the *RfD* is 0.003 for Cr, 0.02 for Ni, 0.04 for Cu, 0.3 for Zn, 0.0003 for As, 0.001 for Cd, and is 0.0035 for Pb [28].

To estimate carcinogenic risk, the Cancer Risk (CR) was computed. The CR is the incremental probability of an individual developing cancer over a lifetime due to exposure to a carcinogenic hazard, and is defined as follows:(4)Risk=ADD×SF
where the cancer slope factor is *SF* ((kg·d)/mg). Of the tracer elements considered in this work, *SF* values for the hand-/object-to-mouth pathway have only been established for As and Cr; these values are 1.0 and 0.5, respectively [28].

## 3. Results and Discussion

### 3.1. The SIR Results

The SIR values, based on the measured tracer element concentrations of food, feces, urine, soil and dust are presented in Table 1 and Figure 1. The median (minimum and maximum) of SIR values were −124.3 (−278.0 to 228.2), −210.2 (−490.1 to 273.8), 27.1 (0.4 to 106.0), −22,532.8 (−29,443.8 to −6215.9), 23.9 (−45.3 to 268.0), 175.3 (−56.4 to 1040.7), 39.2 (−36.4 to 284.0), and −263.2 (−491.4 to 132.3) mg/day for Al, Ba, Ce, Mn, Sc, Ti, Y, and V. When comparing the ratios of the median SIR values obtained for individual elements in this work to the median SIR values reported previously for the same elements, it can be seen that the highest ratio is 1.6 (for Ce) and the lowest is 0 (for Sc). This relatively narrow range indicates that the SIR values obtained in this work are comparable to those reported previously. The coefficients of variation in the SIR values for the tracers examined in this work were not high, with the exception of Al, which was 596.3%. The frequency distribution histograms show few outliers, with most being high values (Appendix A). Experimental factors such as measurement error, source error, and transit time misalignment may lead some of these outliers [29]. Other outliers may be due to the behavior of certain children, such as pica behavior [28] or spending unusually large amounts of time playing in grassland. The medians of the SIR values after removing the high values (see Appendix A and Figure 1) were taken as the final SIR values for children living in the studied e-waste dismantling site in South China.

### 3.2. Soil Ingestion Rate Based on the BTM

The complex metabolism of the human body can lead to different behaviors between the tracers. This is an important reason why different tracers give rise to different SIR values. The reliable estimation of soil absorption for each tracer can improve inter-tracer consistency in soil absorption values [22]. Unfortunately, there are substantial differences between the SIR values estimated by different tracers. Doyle et al. [29] found that this variability may be partly due to measurement error, source error, or transit time misalignment. Regardless of its origin, there is a clear need to identify a reliable tracer for SIR estimation. The BTM method was developed for this purpose [29]. This method depends on the Food-to-Soil (F/S) ratio, which is the ratio from the mass of tracer elements taken from food to the mass of tracer elements in 1g of soil within one day. The most suitable tracer elements are identified by the F/S ratio; the lower an element’s F/S ratio, the closer the estimated SIR value is to the true value [30]. The average F/S ratios determined were 0.000121, 0.001118, 0.000001, 0.012732, 0.000035, 0.000024, 0.000658, and 0.000028 for Al, Ba, Ce, Mn, Sc, Ti, V, and Y in this work. Accordingly, Al, Ce, Sc, Ti, and Y were identified as the best tracer elements. However, the SIR values based on these five tracers still show differences. Both the mean and median SIR values determined using Ti as the tracer were significantly higher than the SIR values obtained for other tracers. 

The estimate based on the best five tracer elements (Al, Ce, Sc, Ti, and Y) was found to be the best approximation of the SIR, i.e., the one expected to be closest to the true value. The SIR determined using the BTM approach are shown in Figure 2 (including the frequency distribution and basic statistical parameters). It is clear that the distribution remained skewed (Figure 3). The SIR observed for the children ranged from −76.8 to 1725.0 mg/day, with 47.9, 148.3, and 383.3 mg/day for median, mean, and 95th percentile values, respectively. These data lie in the reference intervals (Ris; USEPA, 2011), which range from upper (URL) to lower (LRL) reference bounds. The LRL is −112.4–100.9 mg/day, which is considered the lower limit of the 90% confidence interval (CI) of the 2.5th percentile (P2.5), whereas the URL is 516.9–730.2 mg/day, which is considered the upper limit of the 90% CI of the 97.5th percentile (P97.5) [10]. However, since negative SIR values are physically meaningless, the RI of the SIR for children living in e-waste dismantling sites is 0–730.4 mg/day. In this study, 95th percentile values (383.3 mg/g) would be the recommended value.

### 3.3. Comparison of SIR Results

The SIR measurements obtained in this work are compared to those reported in Table 2, which also specifies the regions, ages, and tracer elements considered in each study. In most cases, Al, Ti and Ba emerged as the usual tracer elements for SIR estimation. The mean SIR values obtained for Al range from 2.7 [22] to 154.0 [31] mg/day, while the medians range from −3.3 [22] to 33.3 [32] mg/day. The median SIR value obtained for Al in previous studies is higher than that reported in this study, but the mean is slightly lower. The mean SIR values for Ti reported in the literature range from −544.4 [22] to 3368.0 [25] mg/day, while the medians range from 11.9 [22] to 1861.0 [25] mg/day. Our mean and median SIR values are within these ranges. The SIR means for Ba range from 29.0 [31] to 368.0 [25] mg/day, while the medians range from −19.0 [31] to 394.0 [25] mg/day. The SIR mean that we calculated is in the middle of this range, and the median value is lower than those reported previously, as was also the case for Al.

The Ce- and Sc-based mean SIR values obtained in this work are more similar to those reported previously than the medians, while the median Mn-based SIR values obtained in this study are more widely dispersed than those for the other tracers. Additionally, the median SIR value for Mn was large and negative (−22,532.8 mg/day). The median Mn-based SIR determined by Calabrese et al. was −340.0 mg/day, which is the lowest value reported in the literature [31]. The V-based SIR means range from −182.0 [25] to 456.0 [31] mg/day, and the medians range from −185.0 [25] to 123.0 [31] mg/day. As was the case for Al and Ba, our V-SIR median is low. The Y-based SIR obtained in this study is higher than that in some studies, but the means and medians are similar to those reported previously.

Based on the above, using a single tracer is not enough to obtain an accurate SIR for children. Therefore, in this study, the Best-Tracer Method (BTM) was employed. This method has been used before. Based on the BTM approach, Al, Ti, and La were selected as the best tracers in [22]. Lin et al. [14] selected V, Y, Sc, Ce, and Al as the tracers with the best F/S radio values. We similarly chose the elements with the lowest F/S ratios, namely Al, Ce, Sc, Ti, and Y. The median SIR derived by Calabrese et al. using the BTM approach is negative [22], and the mean is low; this is a consequence of their tracer element selections. The BTM-based SIR values obtained by Lin et al. [14] and in our study are all positive, although the median and mean values obtained in this work are higher than those reported by Lin et al.

It is clear that the SIR values for different regions and ages differ depending on the considered tracer. Indeed, even when the SIR is calculated using the same set of elements using the BTM approach, there are pronounced differences between countries and regions in terms of children’s SIR. It appears that the SIR of children is higher in China in this area (median: 171.5 mg/day) than in America and Canada (median: 100 mg/day and 32.0 mg/day, respectively) [25,28]. This may be attributable to differences in lifestyle. Some researchers observed and disciplined the behavior of children to control for the activity factor. The reference hand-to-mouth and object-to-mouth contact frequencies in children specified in the USEPA exposure factor handbook [28] are 3 times/h inside and 7 times/h outside for hand-to-mouth contact and 1 time/h inside and 1 time/h outside for object-to-mouth contact. The durations of these contacts are not specified. However, the hand-/object-to-mouth contact durations for children in Taiwan are 0.34 and 0.46 min/h, respectively, and the corresponding indoor contact frequencies are 8.91 and 11.39 times/h, respectively [24]. Thus, in theory, the SIR in China would be expected to be higher than in America.

In China, different provinces also have different SIR values. In Taiwan, the average SIR for children aged from 24 to 36 months were 90.7 and 29.8 mg/day in the sand and clay groups, respectively [11], whereas in Hubei, Guangdong, and Gansu provinces these values were 51.7 mg/day [14]. Our study collected children from e-waste dismantling areas in South China, and we estimated that the SIR median was 148.3 mg/day. It is possible that the children studied in Taiwan were younger children who spend most of their time indoors without outside activity, resulting in relatively low SIR values. However, our study and that of Lin et al. [14] focused on children with higher levels of outside activity, who would be expected to have higher SIR values than children who spend most of their time indoors. In general, as reflected in these results, older children (6–17 years old) who have more outside activities have a higher SIR (median: 202.9 mg/day) than younger children (3–6 years old, median: 53.9 mg/day) (*p* < 0.05). No activity data were gathered during this study, but such data would facilitate interpretation of the determined SIR values and would, therefore, be useful to obtain in future.

### 3.4. Health Risk Assessment of Heavy Metals Based on SIR Results

SIR is an important parameter for environmental health risk assessment, not least because hand-/object-to-mouth ingestion is the heavy metal exposure pathway associated with the greatest health risk [33]. Therefore, this study assessed the oral ingestion health risk to children in the studied e-waste dismantling area based on the heavy metal contamination of the area’s soils and the calculated SIR values. In resident and park green areas, the highest concentration of heavy metals is found for Zn and the lowest for Cd. The concentrations of heavy metals in park green areas were slightly higher than that in residential area, which may be attributable to the difference in soil environmental quality management and control policy between these areas. After integrating data from the resident and park green areas, the median concentrations (mg/kg) of Cr, Ni, Cu, Zn, As, Cd, and Pb were 48.8, 63.9, 128, 413, 6.30, 0.513, and 115, respectively. Cr showed higher values than the risk screening values for soil contamination in development land in China (GB36600-2018). That means the living spaces in this area have high levels of Cr pollution.

The corresponding carcinogenic risk, and non-carcinogenic risk of the different heavy metals are shown in Table 3; only Cr and As pose a carcinogenic risk to children. Despite the contamination of the area’s soils, non-carcinogenic risk assessments showed that there was no appreciable oral non-carcinogenic risk to children due to heavy metal contamination. Conversely, *HQ* values between 1 and 10 indicate likely damage to human health [29], and *HQ* values above 10 are associated with serious chronic risks. The 95th *HQ* values based on SIR mean value decreased in the order of Pb > As > Cr > Cu > Ni > Zn > Cd, and all of them were below 1. Even though the results regarding SIR recommended values seem to indicate a low risk, Pb is still the main heavy metal source of non-carcinogenic risk to children in the studied area, and 2.5% of children are exposed to Pb health risks. Zhang et al. [34] also found high concentrations of Pb and Cd in the blood of children in our sampling area, which were much higher than the concentrations in the blood of children in the control area. Due to the influence of Pb on the human nervous system and immune system, this is a matter of concern. The calculated carcinogenic risks for the different heavy metals varied widely. Carcinogenic risk values below 10^−6^ are considered safe; however, the carcinogenic risk due to Cr and As calculated using the SIR recommended value in this study were 3.73 × 10^−5^, and 1.44 × 10^−5^, respectively. In addition, 75% and 50% children were suffering from a high carcinogenic risk caused by Cr and As, respectively.

If SIR is not localized but adopts the recommended value of USEPA (100 mg/day), the health risk to local children will be assessed as being low. Overall, the obtained results suggest that the two elements posing the greatest health risk to children in the studied e-waste dismantling site are Cr and As. Continuous monitoring of their concentration in the area’s soil is required. Otherwise, children are at greater risk of exposure to soil pollution than adults. To reduce the risk of As and Cr, schools should keep desks and teaching aids clean and tidy and urge children to clean up after outdoor activities. Due to the relationship between soil heavy metal pollution and e-waste disposal sites [9], new schools should be built as far away from e-waste disposal sites as possible.

## 4. Conclusions

Al, Ce, Sc, Ti, and Y had the lowest F/S ratios of the elements included in our analysis and were, therefore, better tracer elements for SIR calculation than Ba, Mn, and V. The mean, median, and 95th percentile SIR values calculated based on measurements of these five elements were 148.3, 47.9, and 383.3 mg/day, respectively. Our estimate of children’s SIR in South China was slightly higher than the values reported for America, Canada, and other areas of China. These differences may be due to the regional differences in children’s lifestyles. Thus, the children’s behavior patterns associated with soil intake warrant further investigation. We should also sample a greater number of individuals to improve the accuracy of the subsequent experiments.

The calculated SIR values were used in conjunction with soil pollution data to assess the risks to children’s health due to heavy metal exposure via the hand-/object-to-mouth intake pathway in the studied region. The overall health risk was found to be high. Although the non-carcinogenic risk is within the lowest range specified in the relevant guidelines, the carcinogenic risk for Cr and As was over the acceptable range of below 10^−6^. It should be noted that there are more than 75% children in this area living with a carcinogenic health risk. To our knowledge, this study is the first to apply the tracer mass-balance method to determine the SIR for children living in e-waste dismantling sites in Southern China, and to use the calculated SIR values for health risk assessment in children.

## Figures and Tables

**Figure 1 ijerph-19-07332-f001:**
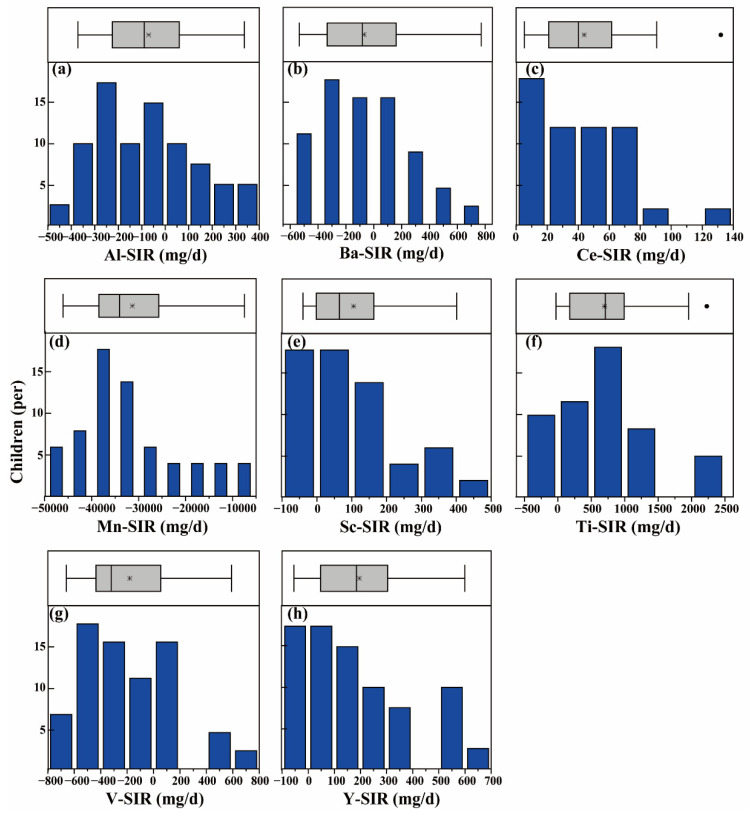
Frequency distribution histogram and outlier box of soil ingestion rate (SIR) based on tracer Al (**a**), Ba (**b**), Ce (**c**), Mn (**d**), Sc (**e**), Ti (**f**), V (**g**), and Y (**h**). The * represents mean values, and dot sign represents outliers.

**Figure 2 ijerph-19-07332-f002:**
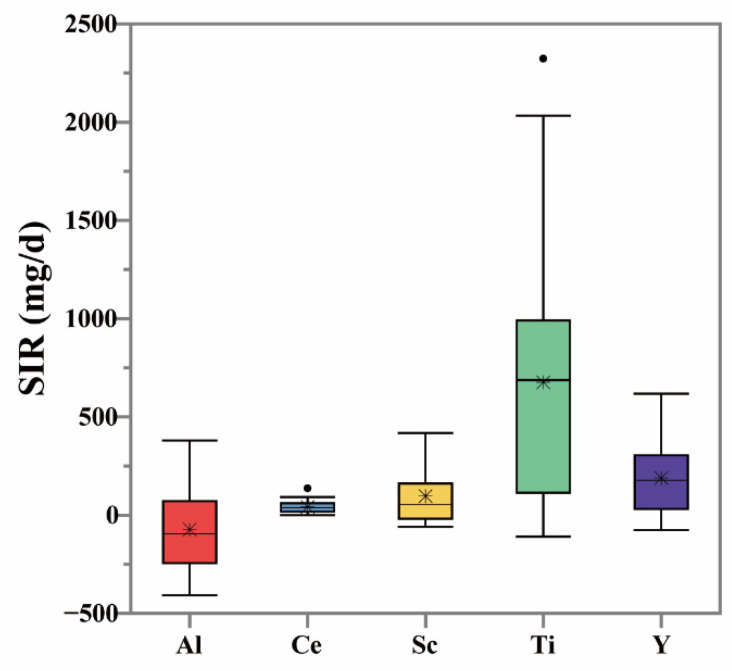
The outlier box plot, minimum, mean, median, and maximum, of SIRs based on the best tracer Al, Ce, Sc, Ti, and Y. The * represents mean values, and dot sign represents outliers.

**Figure 3 ijerph-19-07332-f003:**
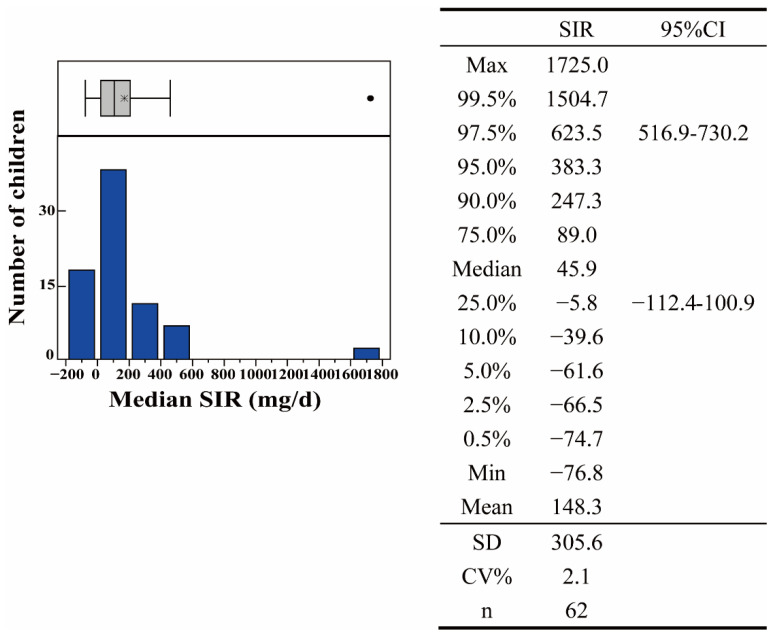
Frequency distribution histogram based on the BTM. All basic statistical parameters of SIR results are listed by the side of histogram. The * represents mean values, and dot sign represents outliers.

**Table 1 ijerph-19-07332-t001:** The result of each day soil ingestion rate (SIR, mg/day) for 62 children based on tracer elements including Al, Ba, Ce, Mn, Sc, Ti, Y, and V.

	Al	Ba	Ce	Mn	Sc	Ti	Y	V
Max	228.2	273.8	106.0	−6215.9	268.0	1040.7	284.0	132.3
99.50%	224.8	267.3	101.0	−6438.2	261.5	988.2	269.5	131.6
97.50%	211.4	241.2	80.8	−7327.3	235.3	778.2	211.2	128.7
95.00%	183.9	213.5	61.4	−9089.1	201.0	579.2	150.1	125.0
90.00%	96.5	156.8	56.4	−14,893.6	128.9	537.3	104.7	99.0
75.00%	−12.8	1.5	48.5	−20,144.1	79.4	439.6	92.9	−32.6
Median	−124.3	−210.2	27.1	−22,532.8	23.9	175.5	39.2	−263.2
25.00%	−183.8	−363.4	13.5	−24,985.0	−15.8	13.7	−1.0	−333.3
10.00%	−233.3	−427.7	6.3	−27,811.6	−25.5	−24.0	−17.4	−414.1
2.50%	−243.7	−437.9	4.4	−28,870.4	−30.4	−30.2	−28.6	−443.8
0.50%	−274.2	−484.3	0.9	−29,391.0	−43.6	−53.5	−35.7	−486.3
Min	−278.0	−490.1	0.4	−29,443.8	−45.3	−56.4	−36.4	−491.4
Mean	112.5	192.6	18.8	4282.5	57.5	227.9	53.0	166.4
SD	138.0	225.9	24.3	5907.3	77.8	274.6	71.0	193.5
CV%	1.2	1.2	1.3	1.4	1.4	1.2	1.3	1.2
n	62	62	62	62	62	62	62	62

**Table 2 ijerph-19-07332-t002:** SIR (mean, median, SD, and SIR recommended, mg/day) comparisons between published studies and this study.

Reference	Age (Years)	n	Region	Tracer Element	Mean	Median	SD	SIR Recommended
Calabrese et al., 1989 [31]	1–4	64	America	Al	154.0	30.0	629.0	154.0
				Ti	170.0	30.0	691.0	
				Ba	29.0	−19.0	868.0	
				Mn	−496.0	−340.0	1974.0	
				V	456.0	123.0	1013.0	
				Y	65.0	11.0	717.0	
Calabrese et al., 1997 [22]	1–4	10	America	Al	2.7	−3.3	95.8	
				Ti	−544.4	11.9	2509.0	
				La	8.6	84.5	1377.2	
				BTM	6.8	−2.4	74.5	
Davis, et al., 2006 [32]	3–8	12	Canada	Al	36.7	33.3	35.4	
				Ti	206.9	46.7	277.5	
Irvine et al., 2014 [25]	Adult	9	Canada	Al	33.0	32.0	55.0	32.0
				Ti	3368.0	1861.0	4277.0	
				Ba	368.0	394.0	725.0	
				Ce	11.0	10.0	34.0	
				La	12.0	11.0	36.0	
				Mn	1363.0	1408.0	5359.0	
				V	−182.0	−185.0	144.0	
				Y	−13.0	1.0	67.0	
Chien et al., 2017 [24]	0.5–3	66	Taiwan, China	Ti	957.1		477.0	
				Si	9.6		19.2	
Lin et al., 2017 [14]	2.5–11.9	177	China	Al	47.7	27.8	59.8	60.8
				Ti	81.9	36.7	177.6	
				Ba	63.1	36.5	125.9	
				Ce	53.5	34.8	48.8	
				Mn	230.8	146.6	617.6	
				Sc	77.7	54.8	68.8	
				V	106.4	92.1	64.6	
				Y	79.8	59.1	68.3	
				BTM	73.5	51.7	63.7	
This study	2–16	61	China	Al	112.5	−124.3	138.0	383.3
				Ba	192.6	−210.2	225.9	
				Ce	18.8	27.1	24.3	
				Mn	4282.5	−22532.8	5907.3	
				Sc	57.5	23.9	77.8	
				Ti	227.9	175.5	274.6	
				V	166.4	−263.2	193.5	
				Y	53.0	39.2	71.0	
				BTM	148.3	47.9	306.5	

**Table 3 ijerph-19-07332-t003:** The estimated children’s health risk assessment results for heavy metal soil pollution in this area based on the data on children’s soil intake obtained from this study.

	Non-Carcinogenic Risk							Carcinogenic Risk (×10^−6^)	
	Cr	Ni	Cu	Zn	As	Cd	Pb	Cr	As
Max	1.42	0.275	0.278	0.120	1.83	0.045	2.87	168	65.0
99.50%	1.24	0.240	0.243	0.105	1.60	0.039	2.50	146	56.7
97.50%	0.512	0.099	0.101	0.043	0.661	0.016	1.04	60.6	23.5
95.00%	0.315	0.061	0.062	0.027	0.406	0.010	0.637	37.3	14.4
90.00%	0.279	0.054	0.055	0.024	0.361	0.009	0.565	33.1	12.8
75.00%	0.209	0.041	0.041	0.018	0.270	0.007	0.423	24.8	9.59
Median	0.087	0.017	0.017	0.007	0.113	0.003	0.177	10.4	4.01
25.00%	0.018	0.003	0.003	0.001	0.023	0.001	0.036	2.091	0.810
10.00%	−0.020	−0.004	−0.004	−0.001	−0.026	−0.001	−0.041	−2.41	−0.934
5.00%	−0.034	−0.007	−0.007	−0.003	−0.043	−0.001	−0.068	−3.98	−1.54
2.50%	−0.044	−0.009	−0.009	−0.004	−0.057	−0.001	−0.089	−5.20	−2.02
0.50%	−0.059	−0.012	−0.012	−0.005	−0.076	−0.002	−0.120	−7.01	−2.72
Min	−0.063	−0.012	−0.012	−0.005	−0.081	−0.002	−0.128	−7.47	−2.89
Mean	0.141	0.027	0.028	0.012	0.182	0.004	0.285	16.7	6.46
SD	0.252	0.049	0.049	0.021	0.325	0.008	0.509	29.8	11.5
n	62	62	62	62	62	62	62	62	62

## Data Availability

The study did not report any data.

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
