# Peer review of "Estimation of Children’s Soil and Dust Ingestion Rates and Health Risk at E-Waste Dismantling Area"

_ijerph, 2022, doi:10.3390/ijerph19127332_

Round 1
Reviewer 1 Report
This study estimated the heavy metal exposure to children through soil and dust Ingestion in electronic waste (e‒waste) dismantling areas. The results provided insights into the varying health risks from different have metals. I suggested the major revision before publication.
- The authors should provide detailed data about the concentration of heavy metals in soil and biotic samples.
- The authors were suggested to introduce the spatial variation of heavy metal in samples, and to analyze the possible factors contributing to the concentration profile of heavy metals in differnt samples.
- The authors should add section(s) for the heavy metal concentrations in biotic samples. As an example, they may analyze the influences of food intake and food intake on heavy metal exposure.
- Lines 116-118: The amounts of samples were confused. Although there were 66 children involved in this study, 66 food samples, 62 feces and 64 urine samples were collected.
- Lines 176-178: Different units between manuscript and SM (figure S1).
- Lines 209-211: Different SIR values between manuscript and figure 1.
- Lines 209 and 214: check the values of Al and Sc, because they were different among discussion, Table 1 and figure 1.
- Lines 218 and 223: check the Figure 3 and figure S1.
- Lines 253 and 377: different values between SIR and figure 3.
- Lines 348-349: correct the order of HQs.
- correct the figure caption in SM.
Author Response
Please see the attachment, including: manuscript, and supporting information.
Point 1、This study estimated the heavy metal exposure to children through soil and dust Ingestion in electronic waste (e-waste) dismantling areas. The results provided insights into the varying health risks of different have metals. I suggested the major revision before publication.
The authors should provide detailed data about the concentration of heavy metals in soil and biotic samples. The authors were suggested to introduce the spatial variation of heavy metal in samples, and to analyze the possible factors contributing to the concentration profile of heavy metals in differnt samples. The authors should add section(s) for the heavy metal concentrations in biotic samples. As an example, they may analyze the influences of food intake and food intake on heavy metal exposure.
Answer: Thank you for your advice. At first, the soil heavy metal data has been added in the supporting information (Table S4 and S5). Besides, our study focuses on estimating the children’s SIRs of from e‒waste disassembly areas by collecting and analyzing selected tracer elements in matched samples of their consumed food, feces, and urine, as well as soil samples from their play areas. We have examined the tracer elements in the food. And when we calculated the SIR, we also used the parameter “Cfood” (equation 1) to analyze the food intake.
In addition, children’s feces and urine belong to biological samples, which is sufficient to calculate the SIR. Application of more biotic samples that you suggested is useful for our study, however, it is hard to get other samples from children, such as blood and tissue.
At last, we also added the heavy metals information in resident or park green areas: “In resident and park green areas, the highest concentration of heavy metals is Zn and the lowest is Cd. The concentrations of heavy metals in park green areas were slightly higher than that in residential area, which may attribute to the difference of soil environmental quality management and control policy between these areas. Integrated data from the resident and park green areas, the median concentrations (mg/kg) of Cr, Ni, Cu, Zn, As, Cd, and Pb were 48.8, 63.9, 128, 413, 6.30, 0.513, and 115, respectively.”
Point 2. Lines 116-118: The amounts of samples were confused. Although there were 66 children involved in this study, 66 food samples, 62 feces and 64 urine samples were collected.
Answer: Thanks for your reminder. In our study, 66 children were taken part in our analysis. But we failed to get all children’s feces and urine sample every day. Thus, we collected sample number was less the involved 66 children.
Point 3. Lines 176-178: Different units between manuscript and SM (figure S1).
Answer: Thank you for your reminder. We have uniformed the units to “(mL/d)”
Point 4. Lines 209-211: Different SIR values between manuscript and figure 1.
Answer: Thank you for your correction. In the manuscript, the data is with outlier. However, we used the data without outlier to construct the figure. At present, the related content in manuscript has been revised to: “The SIR values based on the measured tracer element concentrations of food, feces, urine, soil and dust are presented in Table 1 and Figure 1. The median (minimum and maximum) of SIR values were ‒124.3 (‒278.0 to 228.2), ‒210.2 (‒490.1 to 273.8), 27.1 (0.4 to 106.0), ‒22532.8 (‒29443.8 to ‒6215.9), 23.9 (‒45.3 to 268.0), 175.3 (‒56.4 to 1040.7), 39.2 (‒36.4 to 284.0), and ‒263.2 (‒491.4 to 132.3) mg/d for Al, Ba, Ce, Mn, Sc, Ti, Y, and V.”
Point 5. Lines 209 and 214: check the values of Al and Sc, because they were different among discussion, Table 1 and figure 1.
Answer: Thank you for your correction. In the manuscript, the data is with outlier. However, we used the data without outlier to construct the figure. At present, the related content in manuscript has been replaced to: “The SIR values based on the measured tracer element concentrations of food, feces, urine, soil and dust are presented in Table 1 and Figure 1. The median (minimum and maximum) of SIR values were ‒124.3 (‒278.0 to 228.2), ‒210.2 (‒490.1 to 273.8), 27.1 (0.4 to 106.0), ‒22532.8 (‒29443.8 to ‒6215.9), 23.9 (‒45.3 to 268.0), 175.3 (‒56.4 to 1040.7), 39.2 (‒36.4 to 284.0), and ‒263.2 (‒491.4 to 132.3) mg/d for Al, Ba, Ce, Mn, Sc, Ti, Y, and V.”
Point 6. Lines 218 and 223: check the Figure 3 and figure S1.
Answer: Thank you for your correction. We have reorganized the SM figures and the manuscript content. The modifications are as follows: “The frequency distribution histograms show few outliers, most being high values (Figure S3 (a) and Figure S4). Experimental factors such as measurement error, source error, and transit time misalignment, may lead to some of these outliers. Other outliers may be due to the behavior of certain children, such as pica behavior or spending unusually large amounts of time playing in grassland. The medians of the SIR values after removing the high values (see Figure S3(b) and Figure 1) were taken as the final SIR values for children living in the studied e‒waste dismantling site in South China.”
Point 7. Lines 253 and 377: different values between SIR and figure 3.
Answer: Thank you for your correction. It has been changed to “The LRL is ‒112.4‒100.9 mg/d which is considered as the lower limit of the 90% confidence interval (CI) of the 2.5th percentile (P2.5), whereas the URL is 516.9‒730.2 mg/d, which is considered as the upper limit of the 90% CI of the 97.5th percentile (P97.5). However, since negative SIR values are physically meaningless, the RI of the SIR for children living in e‒waste dismantling sites is 0‒730.4 mg/d. In this study, 95th per-centile values (383.3 mg/g) would be as recommended value.”
Point 8. Lines 348-349: correct the order of HQs.
Answer: Thank you for your correction. It has been changed to “The 95th HQ values based on SIR mean value decreased in the order of Pb > As > Cr > Cu > Ni > Zn > Cd and all of them were below 1.”
Point 9. correct the figure caption in SM.
Answer: Thanks for your suggestion. We have revised all the figure captions in SM.

Reviewer 2 Report
Title: Estimation of Children’s Soil and Dust Ingestion Rates and Health Risk in E Waste Dismantling Area
Authors: Yan Yang * , Mengdi Zhang , Haojia Chen , Zenghua Qi , Chengcheng Liu , Qiang Chen , Tao Long
Revisions Need:
Title- ‘in’ should be used for larger areas, can you add the study locations? Otherwise replace ‘in’ with ‘at’
Rephrase title so as not to repeat ‘and’
Materials and Methods:
I would like to see a better description of the study location. It was only mentioned towards the end – Discussion
Sample Preparation and Instrumental Analysis
-Why were the food samples microwaved? Better explanation needed
- why were the feces samples not microwaved?
- why different processing for different samples? Better explanation needed
Line190 – why was 350d/y taken? Explain for better understanding
Will like to see ethical approval for the project/research? Ethical approval from which institution? Were Consent/assent from participants obtained?
Results and Discussion
Table 1 – is it possible to list/ summarise the data for each day?
Line 294 – what is meaning of BTM
Would prefer re-introduction of the full meaning of acronyms in each sub-section
Line 320 -Also descript the study location in the method section
Line 342- Describe with references the values for carcinogenic risk and non-carcinogenic risks in the methos sections before mentioning it in the discussion section.
Line 346 – What is HQ? Also describe in the methods section
Line 368 – what is the meaning of ‘Whatever’? please rephrase
Will like to see references on the metal concentrations at the e-waste recycling site.
Author Response
Point 1. Title- ‘in’ should be used for larger areas, can you add the study locations? Otherwise replace ‘in’ with ‘at’Rephrase title so as not to repeat ‘and’
Answer: Thanks for your advice, we have made the changes to the title following:” Estimation of Children’s Soil/Dust Ingestion Rates and Health Risk at E‑Waste Dismantling Area”
Point 2. I would like to see a better description of the study location. It was only mentioned towards the end – Discussion
Answer: Thanks for your suggestion. We have introduced the sampling sites and added the following information: “The e-waste dismantling area we studied is an E-Waste Recycling Town located in South China, where the possible human body burden and health consequences of heavy metals exposure have been reported [30].”
- Huang, W.L.; Shi, X.L.; Wu, K.S. Human Body Burden of Heavy Metals and Health Consequences of Pb Exposure in Guiyu, an E-Waste Recycling Town in China. International Journal of Environmental Research and Public Health, 2021, 18(23):12428. https://doi.org/10.3390/ijerph182312428
Point 3. Sample Preparation and Instrumental Analysis
-Why were the food samples microwaved? Better explanation needed
- why were the feces samples not microwaved?
- why different processing for different samples? Better explanation needed
Answer: Thanks for your suggestion, we have sorted out the method part again, and now the modification is as follows: “Feces samples and food samples were pretreated in the same way. Dried samples (1 g) were digested to evaporate at low temperatures of 55°C on a heating plate with 3 mL concentrated nitric acid, 3 mL hydrogen fluoride, and 1 mL perchloric acid (HNO3‒HF‒HClO4). The digestion process was repeated until the sample becomes sticky. Crushed Evaporated samples were microwaved with 2 mL concentrated nitric acid and 3 mL hydrogen peroxide (HNO3‒H2O2). Digestion was performed at 120°C for 5 min, then 160°C for 5 min, and finally 180°C for 15 min. The digested product was diluted to 30 ml with ultrapure water and then stored at 4°C. The supernatant was extracted and analyzed by High Resolution Inductively Coupled Plasma Mass Spectrometry (HR‒ICP‒MS, Nu Attom, England) to determine Mn, Al, Ba, Ti, Ce, V, Sc, and Y.
Urine samples (15 mL) were placed in a digestion tube and digested by microwaving with 2 mL of H2O2 and 3 mL of concentrated HNO3. The conditions of microwave digestion are the same as mentioned above. Finally, the digest was analyzed by HR‒ICP‒MS.
Dried soil and dust samples (0.5 g) were digested as same as feces and food samples on a heating plate by HNO3‒HF‒HClO4. And then samples were microwaved with HNO3‒H2O2. The supernatant was analyzed for Al, Ba, Mn, Ti, and V by Inductively Coupled Plasma Optical Emission Spectroscopy (ICP‒OES, Spectra Arcos SOP, German) which is a method for atomic emission spectroscopy analysis using a light source that generates plasma discharge through high-frequency inductive coupling and by ICP‒MS for Ce, Sc, and Y. In addition, dried soil samples from living spaces (0.5 g) were digested with the same pre-treated and analyzed by ICP‒MS for Pb, As, Cr, Cu, Ni, Cd, and Zn. The standard curve of concentration was used to determine the sample concentration was established by heavy metal standard (all standards were from The Nonferrous Metals Society of China).”
Point 4. Line190 – why was 350d/y taken? Explain for better understanding
Answer: For this question, our answer is following: this data is from the Technical Guidelines for Soil Pollution Risk Assessment of Construction Land in China. The website address of the guidelines is https://www.mee.gov.cn/ywgz/fgbz/bz/bzwb/trhj/201912/W020191224560850148092.pdf
Will like to see ethical approval for the project/research? Ethical approval from which institution? Were Consent/assent from participants obtained?
For this problem, we conducted a questionnaire survey when collecting samples before the experiment, and all the guardians of the sample providers knew and agreed to the experiment.
Point 5. Table 1 – is it possible to list/ summarise the data for each day?
Answer: Thank you for your suggestions. Collecting samples followed the USEPA recommendation of a period of 28 hours from food to feces and urine. Thus, we cannot provide the data for each day. And we also revised the table 1 legend (removing the “each day”).
Point 6. Line 294 – what is meaning of BTM, Would prefer re-introduction of the full meaning of acronyms in each sub-section
Answer: Thank you for your suggestions. For the convenience of you and readers, we have made full descriptions of some acronyms, and the modifications are as follows: “Therefore, in this study, the Best Tracer Method (BTM) was employed. This method has been used before”, “the Hazard Quotients (HQ) values between 1 and 10 indicate likely damage to human health”.
Point 7. Line 320 -Also descript the study location in the method section
Answer: Thank you for your suggestions. We have added the description of sampling site in the materials and methods. We have replied to your first suggestion on Materials and Methods for the specific content to be added.
Point 8. Line 342- Describe with references the values for carcinogenic risk and non-carcinogenic risks in the methos sections before mentioning it in the discussion section.
Answer: We have added the values for carcinogenic risk and non-carcinogenic risks in the support information.
Point 9. Line 346 – What is HQ? Also describe in the methods section
Answer: Thank you for your proposal. For this part, we have added to the methods section, and followed as: “The Hazard Quotients (HQ) is the ratio of daily intake dose of pollutants to reference dose, which is used to characterize the levels of human exposure to non-carcinogenic contaminants through a single pathway which represents the level of non‒carcinogenic risk”.
Point 10. Line 368 – what is the meaning of ‘Whatever’? please rephrase
Answer: Thanks for your advice, for the first question, this part of the ambiguous expression has been modified to “schools should keep desks and teaching AIDs clean and tidy, and urge children to clean up after outdoor activities”
Point 11. Will like to see references on the metal concentrations at the e-waste recycling site.
Answer: Thank you for your advice. At first, the soil heavy metal data has been added in the supporting information (Table S4 and S5). Besides, our study focuses on estimating the children’s SIRs of from e‒waste disassembly areas by collecting and analyzing selected tracer elements in matched samples of their consumed food, feces, and urine, as well as soil samples from their play areas. We have examined the tracer elements in the food. And when we calculated the SIR, we also used the parameter “Cfood” (equation 1) to analyze the food intake.
In addition, children’s feces and urine belong to biological samples, which is sufficient to calculate the SIR. Application of more biotic samples that you suggested is useful for our study, however, it is hard to get other samples from children, such as blood and tissue.
At last, we also added the heavy metals information in resident or park green areas: “In resident and park green areas, the highest concentration of heavy metals is Zn and the lowest is Cd. The concentrations of heavy metals in park green areas were slightly higher than that in residential area, which may attribute to the difference of soil environmental quality management and control policy between these areas. Integrated data from the resident and park green areas, the median concentrations (mg/kg) of Cr, Ni, Cu, Zn, As, Cd, and Pb were 48.8, 63.9, 128, 413, 6.30, 0.513, and 115, respectively.”

Reviewer 3 Report
Dear Authors and Editor,
the work presents interesting results on estimation of the soil ingestion of selected children from e‒waste disassembly areas by collecting and analyzing selected tracer metals in matched samples of their consumed food, feces, and urine, as well as soil samples from their play areas.
I’d suggest some modifications to improve the paper. Please, see my suggestion below:
Page 2, lines 76-78:
You can remove this sentence:
a class of elements in the human body that are not easily absorbed by the human gastrointestinal tract and are also difficult to be transformed into other substances
it’s not necessary
Page 3, lines 99-100:
please, add references for the following information
South China's economic conditions are better than in North China, but the environmental pollution is worse. The site we studied is an e‒waste dismantling area with severe soil pollution that is typical of e‒waste dismantling areas in South China.
Page 3, lines 116 and 120:
Please, add technical details about:
- lyophilization of food samples.
- Freeze-drying conditions of feces.
Page 3, lines 121-124:
Please, add some information:
- How did you collect dust samples?
- How much soil (kg) did you collect for each sample?
- Was the soil quartered?
Page 3, lines 131-151:
Please, add some information:
- How many millilitres of HNO3, H2O2, HF, HClO4 did you use for the digestion procedures (food samples, faces samples, soil and dust samples)?
- Please, use element symbols not the entire name in all the manuscript.
- Temperature of the heating plate for faces sample digestion.
- Specify the mixed standards and certified reference materials used for calculating accuracy of the digestion procedure.
- Which reference solutions were used to evaluate accuracy and precision of ICP-OES and ICP-MS analysis?
- Please, add the instrument model.
Lines 131-135: Please, modify as suggested
Crushed food samples (1 g) were microwaved with concentrated nitric acid and hydrogen peroxide (HNO3‒H2O2). The supernatant was extracted and analyzed by High Resolution Inductively Coupled Plasma Mass Spectrometry (HR‒ICP‒MS) to determine Mn, Al, Ba, Ti, Ce, V, Sc, and Y.
Lines 137-138: Please, modify as suggested
The supernatant was then extracted and analyzed by HR‒ICP‒MS to determine Mn, Al, Ba, Ti, Ce, V, Sc, and Y,
Please, add labels (a, b, c, etc.) in each figure and update the captions. Also in the supplementary. Please, check captions of figures S2 and 3 that are identical.
best regards

Author Response
the work presents interesting results on estimation of the soil ingestion of selected children from e‒waste disassembly areas by collecting and analyzing selected tracer metals in matched samples of their consumed food, feces, and urine, as well as soil samples from their play areas.
I’d suggest some modifications to improve the paper. Please, see my suggestion below:
Point 1. Page 2, lines 76-78: You can remove this sentence: a class of elements in the human body that are not easily absorbed by the human gastrointestinal tract and are also difficult to be transformed into other substances it’s not necessary
Answer: Thanks for your advice, and this explanation has been removed from the manuscript, it is changed to “this method analyzes the concentration of tracer elements in the soil to which children are exposed, the children's intake of food, their excreted feces and urine, and the content of the tracer element in the children’s food, feces, and urine.”
Point 2. Page 3, lines 99-100:
please, add references for the following information
South China's economic conditions are better than in North China, but the environmental pollution is worse. The site we studied is an e‒waste dismantling area with severe soil pollution that is typical of e‒waste dismantling areas in South China.
Answer: Thanks for your advice, and we have added references to support this view, it is added “[30] Huang, W.L.; Shi, X.L.; Wu, K.S. Human Body Burden of Heavy Metals and Health Consequences of Pb Exposure in Guiyu, an E-Waste Recycling Town in China. International Journal of Environmental Research and Public Health, 2021, 18(23):12428. https://doi.org/10.3390/ijerph182312428”
And we changed content as “The e-waste dismantling area we studied is an E-Waste Recycling Town located in South China, where the possible human body burden and health consequences of heavy metals exposure have been reported [30].”
Point 3. Page 3, lines 116 and 120: Please, add technical details about:lyophilization of food samples.
Freeze-drying conditions of feces.
Answer: Thanks for your advice, we have added the specific parameters and modified them as follows:” When the collection was completed, the samples were taken back to the laboratory and stored first in the refrigerator (‒20°C). Then, the feces were freeze‒dried (under vacuum conditions, the vacuum freeze drier temperature is ‒40°C to ‒50°C for 48 h) after measuring the weight with a vacuum freeze dryer.”
Point 4. Page 3, lines 121-124:Please, add some information: How did you collect dust samples?How much soil (kg) did you collect for each sample? Was the soil quartered?
Answer: Thank you for your suggestion. We have added the content we did not mention in this part, but the soil was not quartered, so there is no supplement. Now the modification is as follows: “Collect dust indoors or outdoors by cleaning dust from areas such as tables and windowsills. And 20 g of soil and as much dust (5-20 g) as possible were collected during the sampling process.”
Point 5. Page 3, lines 131-151: Please, add some information:How many millilitres of HNO3, H2O2, HF, HClO4 did you use for the digestion procedures (food samples, faces samples, soil and dust samples)?
Please, use element symbols not the entire name in all the manuscript. Temperature of the heating plate for faces sample digestion. Specify the mixed standards and certified reference materials used for calculating accuracy of the digestion procedure.Which reference solutions were used to evaluate accuracy and precision of ICP-OES and ICP-MS analysis? Please, add the instrument model.
Answer: Thanks for your advice! In view of the imperfection of the method and the defects of expression, we have added and improved this part, and the modification is as follows: “Feces samples and food samples were pre-treated in the same way. Dried samples (1 g) were digested to evaporate at low temperatures of 55°C on a heating plate with 3 mL concentrated nitric acid, 3 mL hydrogen fluoride, and 1 mL perchloric acid (HNO3‒HF‒HClO4). The digestion process is repeated until the sample becomes sticky. Crushed Evaporated samples were microwaved with 2 mL concentrated nitric acid and 3 mL hydrogen peroxide (HNO3‒H2O2). Digestion was performed at 120°C for 5 min, then 160°C for 5 min, and finally 180°C for 15 min. The digested product was diluted to 30 ml with ultrapure water and then stored at 4°C. The supernatant was extracted and analyzed by High Resolution Inductively Coupled Plasma Mass Spectrometry (HR‒ICP‒MS, Nu Attom, England) to determine Mn, Al, Ba, Ti, Ce, V, Sc, and Y.
Urine samples (15 mL) were placed in a digestion tube and digested by microwaving with 2 mL of H2O2 and 3 mL of concentrated HNO3. The conditions of microwave digestion are the same as mentioned above. Finally, the digest was analyzed by HR‒ICP‒MS.
Dried soil and dust samples (0.5 g) were digested as same as feces and food samples on a heating plate by HNO3‒HF‒HClO4. And then samples were microwaved with HNO3‒H2O2. The supernatant was analyzed for Al, Ba, Mn, Ti, and V by Inductively Coupled Plasma Optical Emission Spectroscopy (ICP‒OES, Spectra Arcos SOP, German) which is a method for atomic emission spectroscopy analysis using a light source that generates plasma discharge through high-frequency inductive coupling and by ICP‒MS for Ce, Sc, and Y. In addition, dried soil samples from living spaces (0.5 g) were digested with the same pre-treated and analyzed by ICP‒MS for Pb, As, Cr, Cu, Ni, Cd, and Zn. The standard curve of concentration was used to determine the sample concentration was established by heavy metal standard (all standards were from The Nonferrous Metals Society of China).”
Point 6. Lines 131-135: Please, modify as suggested
Crushed food samples (1 g) were microwaved with concentrated nitric acid and hydrogen peroxide (HNO3‒H2O2). The supernatant was extracted and analyzed by High Resolution Inductively Coupled Plasma Mass Spectrometry (HR‒ICP‒MS) to determine Mn, Al, Ba, Ti, Ce, V, Sc, and Y.
Lines 137-138: Please, modify as suggested
The supernatant was then extracted and analyzed by HR‒ICP‒MS to determine Mn, Al, Ba, Ti, Ce, V, Sc, and Y,
Answer: Thanks for your advice! In view of the imperfection of the method and the defects of expression, we have added and improved this part, and the modification is as follows: “Feces samples and food samples were pre-treated in the same way. Dried samples (1 g) were digested to evaporate at low temperatures of 55°C on a heating plate with 3 mL concentrated nitric acid, 3 mL hydrogen fluoride, and 1 mL perchloric acid (HNO3‒HF‒HClO4). The digestion process is repeated until the sample becomes sticky. Crushed Evaporated samples were microwaved with 2 mL concentrated nitric acid and 3 mL hydrogen peroxide (HNO3‒H2O2). Digestion was performed at 120°C for 5 min, then 160°C for 5 min, and finally 180°C for 15 min. The digested product was diluted to 30 ml with ultrapure water and then stored at 4°C. The supernatant was extracted and analyzed by High Resolution Inductively Coupled Plasma Mass Spectrometry (HR‒ICP‒MS, Nu Attom, England) to determine Mn, Al, Ba, Ti, Ce, V, Sc, and Y.
Urine samples (15 mL) were placed in a digestion tube and digested by microwaving with 2 mL of H2O2 and 3 mL of concentrated HNO3. The conditions of microwave digestion are the same as mentioned above. Finally, the digest was analyzed by HR‒ICP‒MS.
Dried soil and dust samples (0.5 g) were digested as same as feces and food samples on a heating plate by HNO3‒HF‒HClO4. And then samples were microwaved with HNO3‒H2O2. The supernatant was analyzed for Al, Ba, Mn, Ti, and V by Inductively Coupled Plasma Optical Emission Spectroscopy (ICP‒OES, Spectra Arcos SOP, German) which is a method for atomic emission spectroscopy analysis using a light source that generates plasma discharge through high-frequency inductive coupling and by ICP‒MS for Ce, Sc, and Y. In addition, dried soil samples from living spaces (0.5 g) were digested with the same pre-treated and analyzed by ICP‒MS for Pb, As, Cr, Cu, Ni, Cd, and Zn. The standard curve of concentration was used to determine the sample concentration was established by heavy metal standard (all standards were from The Nonferrous Metals Society of China).”
Point 7. Please, add labels (a, b, c, etc.) in each figure and update the captions. Also in the supplementary. Please, check captions of figures S2 and 3 that are identical.
Answer: Thank you for your correction. We have added (a, b, c, ect) to Figure 1 and Figure S1,2,3,4. The title of Figure 1 has been changed to “Figure 1. Frequency distribution histogram and outlier box of soil ingestion rate (SIR) based on tracer Al(a), Ba(b), Ce(c), Mn(d), Sc(e), Ti(f), V(g), and Y(h).” The title of Figure S1 has been changed to “Fig. S1 Histogram and basic statistical parameters of investigated child population age (a), height (b), and weight (c)”. The title of Figure S2 has been changed to “Fig. S2 Histogram and basic statistical parameters of daily food ingestion (a) (g/d, ww) and feces (b) (g/d, dw) and urine (c) (mL/d) excretion for investigated child population.”. The title of Figure S3 has been changed to “Fig. S3 Frequency distribution histogram and outlier box (a) and Frequency distribution histogram and outlier box (b) of soil ingestion rate (SIR) based on Al”. The title of Figure S4 has been changed to “Fig. S4 Frequency distribution histogram and outlier box of soil ingestion rate (SIR) based on tracer Al(a), Ba(b), Ce(c), Mn(d), Sc(e), Ti(f), V(g), and Y(h) separately”.

Round 2
Reviewer 1 Report
I am satisfied with the revision made on the manuscript and would like to recommend the manuscript for publication.